

# Drought cascades across multiple systems in Central Asia identified based on the dynamic space-time motion approach

Lu Tian[1], Markus Disse[1], Jingshui Huang[1]

[1]Chair of Hydrology and River Basin Management, Technical University of Munich, Arcisstrasse 21, 80333 Munich, Germany

*Correspondence to:* Lu Tian (lu.tian@tum.de)

**Abstract.** Drought is typically induced by the extreme water deficit stress that cascades through the atmosphere, hydrosphere, and biosphere. Cascading drought events could cause severe damage in multiple systems. However, identifying cascading drought connections considering the dynamic space-time progression remains challenging, which hinders further exploring the emergent patterns of drought cascades. This study proposes a novel framework for tracking drought cascades across

multiple systems by utilizing dynamic space-time motion similarities. Our investigation focuses on the four primary drought types in Central Asia from 1980 to 2007, namely precipitation (PCP), evapotranspiration (ET), runoff, and root-zone soil moisture (SM), representing the four systems of atmosphere, hydrosphere, biosphere, and soil layer respectively. 503 cascading drought events are identified in this study, including the 261 four-system cascading drought events. Our results show a significant prevalence of the four-system cascading drought pattern in Central Asia with high systematic drought risk, mainly

when seasonal PCP droughts with high severity/intensity and sizeable spatial extent are observed. As for the temporal order in the cascading drought events, ET droughts are likely to occur earlier than Runoff droughts after PCP droughts, and SM droughts are more likely to occur at last, implying the integrated driven effect of the energy-limited and water-limited phases on the drought progression in Central Asia. Our proposed framework could attain precise internal spatial trajectories within each cascading drought event and enable the capture of space-time cascading connections across diverse drought systems and

associated hazards. The identification of cascading drought patterns could provide a systematic understanding of the drought evolution across multiple systems under exacerbated global warming.

## 1 Introduction

Droughts manifest as water deficits in the atmosphere (precipitation), hydrosphere (runoff), biosphere (evapotranspiration), and soil layer (soil moisture) (Orth and Destouni, 2018; Zargar et al., 2011), resulting in a sequential cascade of effects across

interconnected atmospheric-hydrological-biological-agricultural systems.    Anthropogenic global warming amplifies and accelerates this cascading drought chain impacting multiple system sections, exacerbating the underlying systematic risk (Gaupp et al., 2020; Yuan et al., 2023), and increasing uncertainty in predicting drought impacts (Cook et al., 2018; Lehner et al., 2017). Current research primarily focuses on the isolated consequences of precipitation deficits within certain sectors, such as agriculture security, water scarcity, and human health (Fouillet et al., 2006; Jones and van Vliet, 2018; Tuttle and Salvucci,





2017; Vicente-Serrano et al., 2013; Yusa et al., 2015). However, this compartmentalized approach fails to capture the interactions and spatial-temporal dependencies within the entire droughts network, which often results in an underestimation of risk as cascading natural hazards usually cause more severe impacts than any of the single hazard events alone. A comprehensive understanding of drought dynamics evolution across the water cycle necessitates an investigation into the spatiotemporal development of drought cascades. While some knowledge regarding the progression of drought from the

atmosphere (precipitation) to the hydrosphere (runoff), biosphere (evapotranspiration), and soil layer (soil moisture) exists at the catchment scale, this understanding remains limited when considering a larger scale. The complexities arising from interactions among multiple systems and various physical factors make the evolution of drought at a large scale elusive. Expanding our knowledge in this regard would enhance our capacity to predict drought cascades and refine global hydrological and land surface models. This advancement is crucial for better understanding the droughts' evolution in the water cycle,

improving the identification and attribution of cascading droughts, and enhancing global hydrological and land surface models, which is imperative for effectively forecasting and managing the systematic risks associated with drought in the context of global warming.

Conventional methods mainly rely on the point-point or region-region relationship of the drought index to detect the droughts cascade (Apurv et al., 2017; Barker et al., 2016; Farahmand et al., 2021; Geyaert et al., 2018) and do not consider the

dynamic spatial motion over time. For example, Sutanto et al. (2020) consider the sequence of at least two hazards within each geographic grid, uninterrupted by a zero-hazard day, to be a cascading event to study the occurrence pattern of heatwaves, droughts and wildfires at the pan-European scale. Farahmand et al. (2021) targeted the four most significant drought events in the four fixed regions of the US and investigated the cascading drought phenomenon involving precipitation, runoff, soil moisture, streamflow and groundwater through their drought index variation over time at the unified regional scale. However,

the drought index variation in a fixed regional scale cannot reflect the dynamic spatiotemporal nature of drought progression in the real world.

The identification of three-dimensional drought events by characterizing dynamic spatiotemporal continuous motion is introduced in recent study (Diaz et al., 2020; Liu et al., 2020; Zhou et al., 2019), compared to defining the drought by fixing spatial scales, which is able to detect more drought events at moderate and small spatial scale. For example, Yoo et al. (2022)

found that more monthly droughts in 2007 led to long-term droughts in 2008 in Central Asia, which had not been observed before. Therefore, ignoring the temporal variability of drought areas can lead to an underestimation of the probability and occurrence of droughts (Xu et al., 2015), which can further lead to failure to detect cascading phenomena and miscalculation of large-scale droughts cascade. Some studies have started to introduce dynamic spatiotemporal movement in identifying the connection between two types of droughts (Jiang et al., 2023; Liu et al., 2019a). For instance, Liu et al. (2019a) paired two

types of drought events by setting the temporal-spatial overlapping threshold to establish the causal mechanism between them at the catchment scale. Jiang et al. (2023) adopted this approach and used machine learning methods further to examine the propagation possibility of precipitation to ecological droughts. In the real world, the complexity of coupled and interdependent



feedback loops across multiple systems leads to variations in droughts at different spatiotemporal scales, resulting in overlaps that cannot be predetermined within the drought cascade.

Additionally, the basic idea in existing research is to establish the pointwise correlation between the two types of droughts using the standardized hydrological drought index and meteorological droughts at various long-term time scales, by which the time delay among the different droughts could be recognized (Barella-Ortiz and Quintana-Segui, 2019; Barker et al., 2016; Geyaert et al., 2018; Van Loon et al., 2012). However, global warming has sped up the drought intensification rates and triggered the global transition from conventional droughts to more flash droughts (Yuan et al., 2023). The coarse temporal
resolution is inadequate for understanding the wicked problems of rapid temporal-spatial transitions during cascading processes that unfold over short-term scales (Bachmair et al., 2015), particularly for flash droughts.

Based on above, this study introduces a novel approach that considers the nature of the dynamic space-time progression in complex drought systems. An identification technique based on spatiotemporal motion similarity was adopted to determine and characterize cascading drought events across multiple systems. Comparing to existing methods, this approach could detect
the dynamic space-time cascading connection, utilizes high temporal resolution data and eliminates the need for predefined spatiotemporal overlap thresholds. Central Asia (CA) is a typical arid and semi-arid region worldwide. Despite a long history of frequent drought occurrences, the emergent pattern of the drought cascades in CA has not been studied yet. To this end, we employed the proposed method to identify the emergent pattern of the drought cascades in CA and explore their features, including the number of systems involved, temporal order, and total severity. Besides, four characteristics (intensity, severity,
duration, and area) of the single drought event are investigated to explore the drought evolution during the cascading progression. The findings of this study are anticipated to enhance our comprehension of the systematic evolution of drought in arid and semi-arid regions, thereby contributing scientific evidence for the prediction of drought cascades in the context of global warming.





## 2 Material & Methods

**2.1 Study area**

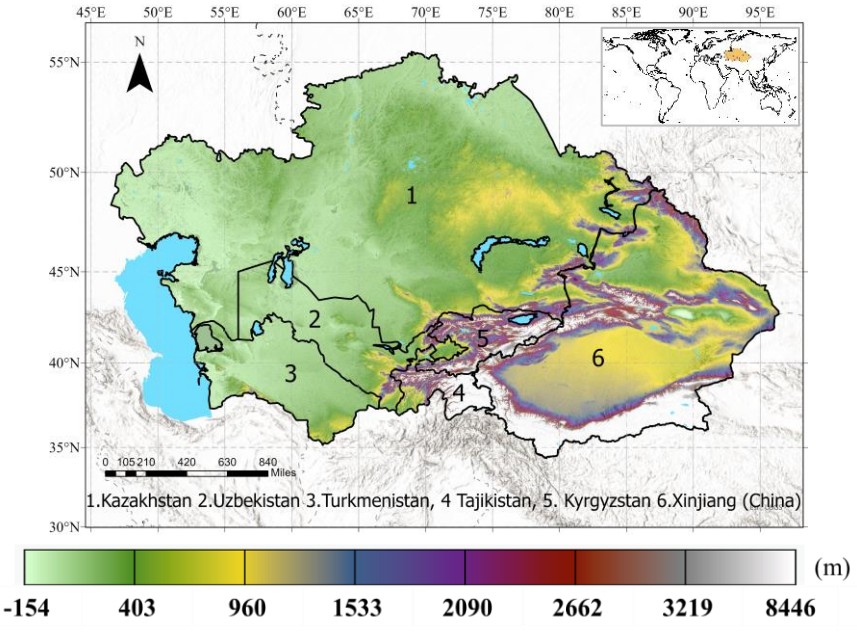

**Figure 1. The topography of Central Asia.**

We used the definition of the geographical scope of Central Asia by (Hu et al., 2018) as the spatial domain of this study area, namely the north-western regions of China (Xinjiang region), the five countries of the former Soviet Union: Kazakhstan,

Kyrgyzstan, Tajikistan, Turkmenistan, and Uzbekistan. Located in the centre of Eurasia and far from the ocean, CA is a typical arid and semi-arid region, often experiencing long-term rainfall deficits and having a variable topography. Figure 1 shows that the topography descends from the eastern mountain ranges to the western steppe region. The Tianshan Mountains, the Pamir Plateau, and the Tarim Basin dominate the terrain and contribute to the region of the eastern alpine and arid desert climates (Xinjiang, Kyrgyzstan, and Tajikistan). The western and central areas are dominated by lower elevations (e.g., Caspian

depression), characterized by extensive temperate grassland and shrubland (steppe) zones. This topography results in significant temperature and precipitation gradients from north to south and lowlands to mountains (Beck et al., 2018). The complex interplay between the variable topography and frequent precipitation deficits (Hu et al., 2018; Spinoni et al., 2019) leads to a fragile state within the terrestrial environment, creating conditions conducive to drought events (Guo et al., 2019).

**2.2 Datasets**

In this study, precipitation (PCP) droughts, evapotranspiration (ET) droughts, Runoff droughts, and root zone soil moisture (SM) droughts in the water cycle are investigated to identify and characterise the drought cascades across four types of systems, namely the atmosphere, hydrosphere, biosphere, and soil layer. The datasets used in this study cover the time window from



January 1st, 1980, to December 31st, 2007. The datasets used in this study were standardized to a daily resolution and a spatial resolution of 0.25 degrees. The details of the dataset information are provided below.

**Precipitation:** APHRODITE (Asian Precipitation—Highly Resolved Observational Data Integration Towards Evaluation of Water Resources) is the long-term continental-scale daily product with a high spatial resolution for the Asian region. This dataset contains interpolated data from more than 6000 rain-gauge observations in Asia by the distance-weighting method (Yatagai et al., 2012). APHRODITE has been widely used as the benchmark dataset to evaluate the performance of different remote sensing products in various regions throughout Asia (Guo et al., 2017; Guo et al., 2015; Iqbal and Athar, 2018;
Jamandre and Narisma, 2013).

    **Evapotranspiration:** Evaporative Deficit, as the result of actual ET (AET) subtracting potential ET (PET), is the standard direct input of the ET-induced drought index. The potential and actual ET derive from the GLEAM v3.5a project (Global Land Evaporation Amsterdam Model Version 3), have been extensively used to identify drought due to its high temporal-spatial resolution and robust performance (Jiang et al., 2021; Peng et al., 2020). GLEAM calculates the PET with the Priestley-Taylor
equation using ERA5 net radiation and air temperature (Miralles et al., 2011b). AET is converted from PET using a multi-layer water-balance algorithm by considering net precipitation (precipitation minus interception loss) and snowmelt (Martens et al., 2017). This dataset has been validated through the FLUXNET global network of micrometeorological flux measurements under various climatic and vegetation situations (Miralles et al., 2011a).

    **Root Zone Soil Moisture:** Root zone soil moisture is obtained from the GLDASv2.0-Noah dataset ensembles (Sheffield
et al., 2006). The daily soil moisture used in this study is the average value of raw data at a 3-hour resolution each day. The soil module of the GLDAS-Noah model comprises four layers, namely 0-10, 10-40, 40-100, and 100-200 cm. The depth of the root zone area depends on the vegetation type. Among the drought recognition ability of remote sensing products, the GLDAS-Noah dataset demonstrated greater capability in detecting drought occurrence over the vegetated region and bare area (Liu et al., 2019; Ma et al., 2019).

**Runoff:** Runoff is the average value of total surface runoff ($Q_{total}$ = surface runoff + subsurface runoff) from seven state-of-the-art global hydrological models' outputs (DBH, H08, LPJmL, MATSIRO, PCR-GLOBWB, VIC, and WaterGAP 2). These data are from ISIMIP2a (Inter-Sectoral Impact Model Intercomparison Project) and forced by the meteorological data from the Watch Forcing Data ERA-Interim (WFDEI) under the 'varsoc' scenario, often used as the validation dataset to evaluate the uncertainty of novel historical runoff reanalysis datasets (Ghiggi et al., 2019). The runoff data at 0.5° resolution was
interpolated into 0.25° by nearest neighbour method in this study.

### 2.3 Methodology

As precipitation deficits are generally considered as the initiation of droughts, PCP droughts are considered the commencement of a cascading drought event in this study. Here, a drought event is defined by space-time continuum movement to account for the spatial variations in drought development. This method could identify more drought events of large and small spatial scale





than conventional determining methods (Yoo et al., 2022). Additionally, the method was established on a daily time scale given the monthly time scale is traditionally too coarse to capture the rapidly intensifying drought events (e.g., flash droughts). Dynamic temporal motion techniques allow the identification of significantly more ET/runoff/SM drought candidates following a PCP drought than conventional methods. To determine and characterize a drought cascading event, the approach proposed here is summarised in three parts (Fig. 2):




**Figure.2. Schematic diagram of methodology detecting the multiple-system cascading event from the space-time dimension**



### 2.3.1 Identifying drought events based on the dynamic space-time motion

#### 2.3.1.1 Drought pixel

To determine drought pixels, the Standardized Drought Analysis Toolbox (SDAT) were used (Farahmand and AghaKouchak,
2015). The standardized indices (SIs) of one hydrological component were calculated at a daily time scale with 30 days
accumulation period. The sum of the corresponding variable for 30 days was compared with the same day on a one-year return
period throughout the study period to get the ranking position $i$. Then, the $i$ value and overall sample size $n$ are the input to the
formulation. Four drought indices are employed in this study to represent the crucial drought processes in the water cycle,
namely the Standardized Precipitation Index (SPI) (McKee, 1993), the Standardized Evapotranspiration Deficit Index (SEDI)
(Vicente-Serrano et al., 2018), the Standardized Soil Moisture Index (SSmI) (AghaKouchak, 2014), and the Standardized
Runoff Index (SRI) (Shukla and Wood, 2008). The Weibull non-exceedance probability calculated the centre of probability
mass of multiple zeros results to compensate for the non-negligible error from long-zero rainfall and runoff period and get
more statistically meaningful SI values. To this end, the SPI and SRI are calculated by Weibull non-exceedance probability
(Stagge et al., 2015) and empirical Gringorten plotting position (Farahmand and AghaKouchak, 2015). The SEDI and SSmI
only are computed by the empirical Gringorten plotting position. Finally, we could get four kinds of SI$_s$ for each cell daily
throughout the study period. The cells with the values lower than -1 are defined as drought pixels (Peng et al., 2020).

#### 2.3.1.2 Drought event

Drought cells (defined by SPI) with sharing sides within one day were clustered together. We used 10 pixels as the minimum
spatial threshold for a drought cluster in this study (Andreadis et al., 2005). Most drought clusters are irregular polygons.

After all drought clusters were generated throughout the study area over the study period, the drought clusters on each day
were linked together to form a drought event. The linking rule is to check if there is an overlapping pixel at adjacent time steps.
This procedure ended when no following spatial overlapping cluster existed on the next time step. The tree traversal was
utilized to ensure a daily cluster only belong to one drought event.

All clusters would be classified as numerous individual assemblies after being traversed. The time step spanned by each
assembly was characterized as duration. Considering the recent developments of flash droughts across the world (Basara et al.,
2019; Christian et al., 2021), we adopt 15 days as the minimum duration threshold following the general definition of flash
drought that lasts at least two weeks (Pendergrass et al., 2020), to incorporate the monthly and sub-monthly droughts. Hence
the independent assemblies over 15 days were kept and viewed as separate drought events. This step allows us to identify each
drought event's onset, end time, and track its spatial drought movement path (see Fig.2).

The same procedure as for the PCP drought event was repeated to identify ET/Runoff/SM drought candidates but within the
PCP area. Furthermore, existing research emphasizes that the propagation of drought impacts from precipitation deficits to
hydrological droughts has been proven to be less than 12 months (Ding et al., 2021). To this end, the threshold from the PCP





drought onset day to 365 days was applied as the temporal boundary in assessing the sequential cascading relationship of ET/runoff/SM droughts.

### 2.3.2 Determining cascading drought events based on the similarity of the space-time motion

This section aims to identify the cascading drought event for a PCP drought from multiple candidates of ET, Runoff, and SM droughts based on the similarity rule of the space-time motion. The steps outlined in this section aim for a single PCP drought event, which will be replicated independently for all PCP droughts to identify all the associated cascading drought events. As the selection process is identical for all ET, runoff, and SM drought candidates, the "$D_{candidate}$" will be used collectively to refer

to all three types of drought candidates in the following description. The two criteria employed in determining the cascading drought events are elaborated below. Based on two criterions, the ET / Runoff / SM drought are determined for each PCP drought to form the multi-system cascading drought event. Notably, there is only one drought event for each type in each cascading drought event.

#### 2.3.2.1 Interrelationships between time series of drought event areas

The first criterion is the correlation between the spatial area changing curve across time of multiple $D_{candidate}$ $\{y'_{Dcandidate}\}$ and their corresponding PCP drought event $\{y_{PCP}\}$. The time series in the real world usually presents autocorrelation and high-degree nonstationary, which could cause spurious correlations. Therefore, the correlation in this study is estimated by Detrended Cross-Correlation Analysis (DCCA) (Podobnik and Stanley, 2008; Zebende, 2011) and Detrended Fluctuation Analysis (DFA) (Peng et al., 1994; Peng et al., 1992). It is particularly useful when analysing nonstationary and autocorrelated

data.

**Step 1:**Compute two integrated signals of two cross-correlated time series with equal length N: $\{y_i\}$ and $\{y'_i\}$. k =1, 2, 3… N.

$$Y_k \equiv \sum_{i=1}^{k} y_i \qquad (1)$$

$$Y_k' \equiv \sum_{i=1}^{k} y_i' \qquad (2)$$

**Step 2:** Divide each time series into N-n overlapping boxes containing n+1 values. The ordinate of a linear least-squares fit of each time series segment is the 'local trend': $\tilde{Y}_{k,i}$ and $\tilde{Y}_{k,i}'$ $(i \leq k \leq i + n)$. The difference between the original walk and the local trend is defined as the 'detrended walk.'

**Step 3:** The integrated signal $Y_k$ is detrended by subtracting the local trend $\tilde{Y}_{k,i}$ in each box.

$$f_{DCCA}^2(n, i) \equiv \frac{1}{n-1} \sum_{k=i}^{i+n} (Y_k - \tilde{Y}_{k,i})(Y_k' - \tilde{Y}_{k,i}') \qquad (3)$$

**Step 4:** The $F_{DCCA}^2(n)$ for the time signal is counted by:



$$F^2_{DCCA}(n, i) \equiv \frac{1}{N-n}\sum_{i=1}^{N-n} f^2_{DCCA}(n, i) \tag{4}$$

The indicator, DCCA cross-correlation coefficient ($\rho_{DCCA}$), is defined as the ratio between the detrended covariance function $F_{DCCA}$ and the detrended variance function $F_{DFA}$ (see Eq. (5)). $\rho_{DCCA}$ is a dimensionless coefficient that ranges between $-1 \leq \rho_{DCCA} \leq 1$. When $\rho_{DCCA} = 1$ means perfect cross-correlation; $\rho_{DCCA} = 0$ means no cross-correlation; $\rho_{DCCA} = -1$ means perfect anti cross-correlation. In this study, we believe that the effective correlation exists when the coefficients higher than 0.5 (Zebende, 2011). The SI$_{candidate}$ with $\rho_{DCCA}$ below 0.5 in the correlation coefficients assemblies will be removed. This step determines the system number of the final cascading drought event. For example, if there are only ET drought candidates is correlated with PCP droughts with $\rho_{DCCA} > 0.5$ and no SM and Runoff droughts candidates meet this threshold, the cascading drought event is the two-system cascading drought event as there is only two types of droughts.

$$\rho_{DCCA} \equiv \frac{F^2_{DCCA}}{F_{DFA}\{y_{PCP}\} F_{DFA}\{y'_{Dcandidate}\}} \tag{5}$$

### 2.3.2.2 Spatial motion direction of drought events

After filtering by the first criterion, the second criterion for cascading connections is the difference in dominated movement direction between PCP and multiple D$_{candidate}$. Dominant movement direction was recognized as the two geographical angles reflecting in drought movement trajectory: 1) the geographical angle from the onset cluster centre to the peak cluster centre; 2) the second is from the peak cluster centre to the termination cluster centre. The centre was calculated by the arithmetic mean of the longitude and latitude of each drought cluster. The schematic of the trajectory of a drought event is displayed in Fig.2. The drought event, with the minimum direction difference relative to the PCP drought event, is identified to form the final cascading event.

### 2.3.3 Features of cascading drought events

First, four characteristics of each single drought event within a cascading event are evaluated, including 1) duration, 2) intensity, 3) severity, and 4) area. The temporal span of an event was defined as the duration of a drought event $T$. Intensity and severity were defined in Eq. (7) and Eq. (8). Area is the number of all pixels of daily drought clusters but overlapped areas of clusters at different day only count once.

$$I_j(t) = SI \tag{6}$$

$$I = \frac{1}{T}\sum_{t=1}^{T} \frac{\sum_{j=1}^{N} I_j(t)}{N} \tag{7}$$

$$S = \sum_{t=1}^{T}\sum_{j=1}^{N} I_j(t) \tag{8}$$





where $I_j(t)$ is the intensity at day t in grid cell j, namely standardized drought index calculated in section 2.3.1.1. $N$ is the number of grid cells in the cluster at time t. $I$ denotes the intensity of a drought event, which is the mean intensity of all drought cells during the drought's duration. $S$ is the drought severity, the accumulation of $I_j(t)$ along with duration $T$ over area.

Second, the number of systems involved, temporal order, and total severity were used to categorize these cascading drought events in this study. All cascading drought event with the same number of systems involved and temporal order were categorized as the same cascading drought pattern. The total occurrence of cascading patterns was calculated as the event number per cascading pattern divided by the total number of cascading drought events (508) from 1980–2007 across Central Asia.

**Table 1 Cascading drought patterns considered for analysis in Central Asia**

| Number | Combinations of single droughts | Number of systems involved |
|---|---|---|
| 0 | PCP | No cascading drought |
| 1 | PCP + ET | |
| 2 | PCP + Runoff | Two-system cascading drought pattern |
| 3 | PCP + SM | |
| 4 | PCP + ET+ Runoff | |
| 5 | PCP + Runoff +ET | |
| 6 | PCP + ET + SM | |
| 7 | PCP + SM + ET | Three-system cascading drought pattern |
| 8 | PCP + Runoff + SM | |
| 9 | PCP + SM + Runoff | |
| 10 | PCP + ET + Runoff + SM | |
| 11 | PCP + ET + SM + Runoff | |
| 12 | PCP + Runoff + ET + SM | |
| 13 | PCP + Runoff + SM + ET | Four-system cascading drought pattern |
| 14 | PCP + SM + ET + Runoff | |
| 15 | PCP + SM + Runoff + ET | |

The 15 cascading patterns in Table 1 were identified in this study. For example, PCP+ET+SM denotes the order of occurrence starting with precipitation droughts, followed by the ET droughts, and ending with SM drought event. Furthermore, the number of single drought events in a cascading event determines the system number, which also denotes the systems involved in the cascading drought events. For example, PCP + SM + ET + Runoff belongs to the four-system cascading drought event, as it includes four single drought events associated with four systems. The temporal order is predominantly determined 240 by the time delay (TD) between PCP droughts and ET/Runoff/SM droughts. It is computed by measuring the actual time lag between similar spatial and temporal movement components (namely $p_{DCCA} > 0.5$) of PCP droughts and the corresponding runoff / ET / SM droughts (Fig. 2).



In addition, we employed the total severity of the cascading drought event to evaluate the systematic severity. The total
severity ($S_{total}$) of the cascading drought event was the cumulative value of the severity of all included single drought events;
see Eq. (9), where $p$ is the number of single drought events within a cascading drought event. The number of systems involved
is the upper limit of individual drought event number (p) in each cascading drought event.

$$S_{total} = \sum_{p=1}^{System} S_p \qquad (9)$$

## 3 Result

### 3.1    Patterns and occurrence of cascading drought events

This study identified 503 cascading drought events. Among these, four-system cascading drought pattern (51.38%) occurred
more frequently than the three-system (37.99%) and two-system (9.64%) cascading drought pattern across the study region.
This higher prevalence of four-system cascading patterns suggests that PCP, ET, runoff, and SM droughts tend to occur in a
consecutive manner instead of in isolation. Notably, the cascading drought patterns that occurred with a frequency greater than
10% were PCP + ET + Runoff (17.52%), PCP + Runoff + ET (16.93%), and PCP + ET + Runoff + SM (12.40%). In contrast,
the PCP + SM cascading pattern, which is associated with root zone soil moisture, was the least frequent cascading pattern,
accounting for less than 0.4% during the study period. ET droughts exhibited a shorter average time delay, with a higher
concentration of data points in the lower quartile of the violin boxplot (Fig. S1). In contrast, SM droughts have a longer average
time delay, with a greater concentration of data points in the upper quartile of the violin boxplot. This evidence suggests that
ET droughts occur predominantly as the initial system of the drought cascade following PCP droughts, while Runoff and SM
droughts tend to follow sequentially as the second and third positions, respectively.





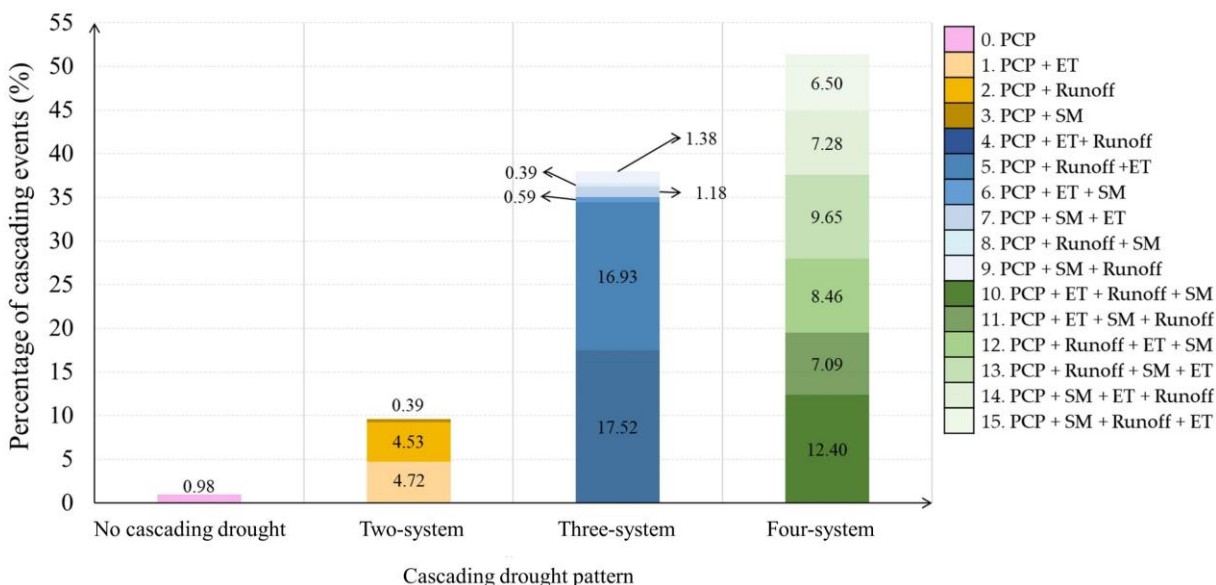

**Figure 3. Total occurrence of all cascading drought patterns**

## 3.2 Characteristic evolution within the cascading drought event

Notable characteristic evolution in duration, area, and severity is observed among PCP, ET, Runoff, and SM droughts during the cascading process. The duration of droughts was significantly extended when the droughts progressed from the atmosphere to the terrestrial system. Specifically, the duration of PCP droughts is the shortest among the four types of droughts. In contrast, SM droughts involving the root zone soil layer exhibit the most extended duration (Fig.4a) in all cascading drought patterns. For the ET, Runoff, and SM droughts occurring in the terrestrial system, the area of ET droughts is significantly greater than

that of Runoff droughts. The affected area of SM drought is consistently the smallest among all cascading patterns (Fig.4b). Regarding severity (Fig.4c), SM droughts exhibit the lowest average severity among the three types of cascading droughts occurring on the land surface in three-system and four-system cascading patterns. But several extremely severe SM droughts exceed the upper threshold of severity for ET and Runoff droughts.

The upper limits regarding the duration, area, and severity of PCP, ET, Runoff, and SM droughts are significantly higher in

four-system cascading patterns compared to the same drought type in two-system and three-system cascading patterns (Fig. 4), indicating that the high-system cascading pattern consists of more pronounced drought events. For example, all droughts in two-system cascading patterns last less than 365 days, while in three-system and four-system cascading patterns, drought events persisting for more than 365 days are much more (Fig. 4a). These findings suggest that when single long-term droughts with markedly severe distribution over large areas are observed, the high-system cascading drought event will likely to

accompany.



**Figure 4: The duration, area, and severity of drought events in the cascading drought patterns.**

### 3.3    The 9 most severe cascading drought events

Based on the total severity, the 3 most severe cascading drought events for different numbers of systems invovled are identified
and the corresponding drought characteristics are listed in Table 2. As shown, the total severity of three top cascading events
in the two-system cascade are from -11.46 to -32.87*$10^6$ Day*$km^2$, remarkably lower than that in three-system (-78.15 to -
127.84 *$10^6$ Day*$km^2$) and four-system cascading event (from -2038 to -925.78*$10^6$ Day*$km^2$). The higher total severity of
cascading events with more systems involved suggests that the increased systematic risk when the cascading drought event
involves more systems.





The most severe cascading drought event is the PCP+Runoff+ET+SM (No.7) in the four-system cascading pattern. This cascading event commenced from the PCP drought on 09-Dec-2006 and terminated by the end of PCP droughts on 29-Dec-2007, lasting for over 12 months and spanning 99.8% of the study area. During the same period, the Runoff drought emerged on 01-Jan-2007 and terminated on 13-May-2007, lasting 133 days. The ET drought followed the Runoff drought and appeared in the summer lasting from 24-Jun-2007 to 29-Jul-2007. The SM drought occurred at the latest and persisted for 50 days with

the smallest affected area. The severe drought event that occurred in 2007 coincided with the findings of Yoo et al. (Yoo et al., 2022). The Food and Agriculture Organization (FAO) has also documented substantial damage to the agriculture sector due to the drought event affecting Turkmenistan, Tajikistan, and Kyrgyzstan in 2007 (Patrick, 2017).

As the cascading connection in this study is identified by the spatial-temporal dynamic movement rather than fixed spatiotemporal overlapping thresholds, there is not always a temporal overlapping pattern in all cascading drought events. For

instance, in the No.2 cascading drought event, the PCP drought lingers from 19-Jul-2002 to 06-Aug-2002, while the ET drought commences on 26-Jul-2002, during the middle of the PCP's persistence period. In contrast, in the No.5 cascading drought event, the PCP drought occurred during the spring of 1997, lasting from 22-Mar-1997 to 09-Apr-1997, and does not exhibit temporal overlap with the subsequent SM drought from 22-June-1997 to 17-Jul-1997.

**Table 2. Top 3 cascading drought events in two-system, three-system and four-system cascading drought pattern according to the**

**total severity**

| Type | Number | Pattern | Drought type | Onset | Termination | Severity ($10^6$ Day*km$^2$) | Area ($10^5$ km$^2$) | Duration (Day) | Maximum intensity | Total severity ($10^6$ Day*km$^2$) |
|---|---|---|---|---|---|---|---|---|---|---|
| Two-system cascading pattern | 1 | PCP+ET | PCP | '15-Aug-2001' | '19-Sep-2001' | -2.45 | 2.34 | 36 | -1.4 | -32.87 |
| | | | ET | '03-Jan-2002' | '13-May-2002' | -30.43 | 2.28 | 131 | -2.06 | |
| | 2 | PCP+ET | PCP | '19-Jul-2002' | '06-Aug-2002' | -1.7 | 1.55 | 19 | -1.77 | -13.18 |
| | | | ET | '26-Jul-2002' | '13-Dec-2002' | -11.48 | 1.54 | 141 | -1.82 | |
| | 3 | PCP+Runoff | PCP | '04-Oct-1991' | '26-Oct-1991' | -3.53 | 2.58 | 23 | -1.53 | -11.46 |
| | | | Runoff | '26-Apr-1992' | '09-Aug-1992' | -7.93 | 1.71 | 106 | -1.66 | |
| Three-system cascading pattern | 4 | PCP+ET+Runoff | PCP | '27-May-1983' | '21-Jul-1983' | -5.88 | 6.88 | 56 | -1.53 | -127.84 |
| | | | ET | '23-Dec-1983' | '20-Jan-1984' | -0.06 | 0.06 | 29 | -1.58 | |
| | | | Runoff | '10-Jan-1984' | '09-Feb-1985' | -121.89 | 6.55 | 396 | -1.84 | |
| | 5 | PCP+SM+Runoff | PCP | '22-Mar-1997' | '09-Apr-1997' | -3.95 | 3.05 | 19 | -1.76 | -119.16 |
| | | | SM | '22-Jun-1997' | '17-Jul-1997' | -0.36 | 0.15 | 26 | -1.3 | |
| | | | Runoff | '01-Apr-1997' | '24-Mar-1998' | -114.84 | 2.98 | 358 | -1.99 | |
| | 6 | PCP+Runoff+ET | PCP | '18-Apr-2005' | '03-May-2005' | -4.51 | 8.05 | 16 | -1.45 | -78.15 |
| | | | Runoff | '16-Aug-2005' | '03-Sep-2005' | -0.38 | 0.27 | 19 | -1.59 | |
| | | | ET | '19-Sep-2005' | '02-Mar-2006' | -73.26 | 7.03 | 165 | -1.82 | |
| | 7 | | PCP | '09-Dec-2006' | '29-Dec-2007' | -2022.85 | 63.09 | 386 | -1.87 | -2038.82 |





| | | | | | | | | | |
|---|---|---|---|---|---|---|---|---|---|
| | PCP+Runoff+ET+SM | Runoff | '01-Jan-2007' | '13-May-2007' | -12.83 | 4.69 | 133 | -2.04 | |
| | | ET | '24-Jun-2007' | '29-Jul-2007' | -2.24 | 1.03 | 36 | -1.72 | |
| | | SM | '10-Nov-2007' | '29-Dec-2007' | -0.89 | 0.16 | 50 | -1.21 | |
| Four-system cascading pattern | 8 PCP+SM+Runof+ET | PCP | '25-Jan-1991' | '10-Aug-1991' | -314.02 | 51.83 | 198 | -1.75 | -1336.31 |
| | | SM | '07-Jul-1991' | '15-Mar-1998' | -1012.44 | 21.32 | 2442 | -1.7 | |
| | | Runoff | '21-Oct-1991' | '21-Dec-1991' | -2.05 | 0.61 | 62 | -1.74 | |
| | | ET | '03-Jan-1992' | '12-Jun-1992' | -7.8 | 1.34 | 161 | -2.04 | |
| | 9 PCP+SM+Runoff+ET | PCP | '10-Apr-1984' | '16-Dec-1984' | -406.59 | 61.74 | 251 | -1.84 | -925.78 |
| | | SM | '17-May-1984' | '09-Jun-1984' | -0.26 | 0.08 | 24 | -1.55 | |
| | | Runoff | '12-Jul-1984' | '06-Aug-1984' | -0.72 | 0.35 | 26 | -1.39 | |
| | | ET | '24-Oct-1984' | '18-Oct-1986' | -518.21 | 52.92 | 725 | -1.99 | |

## 3.4 Dynamic temporal-spatial progression of all single droughts in a cascading drought event

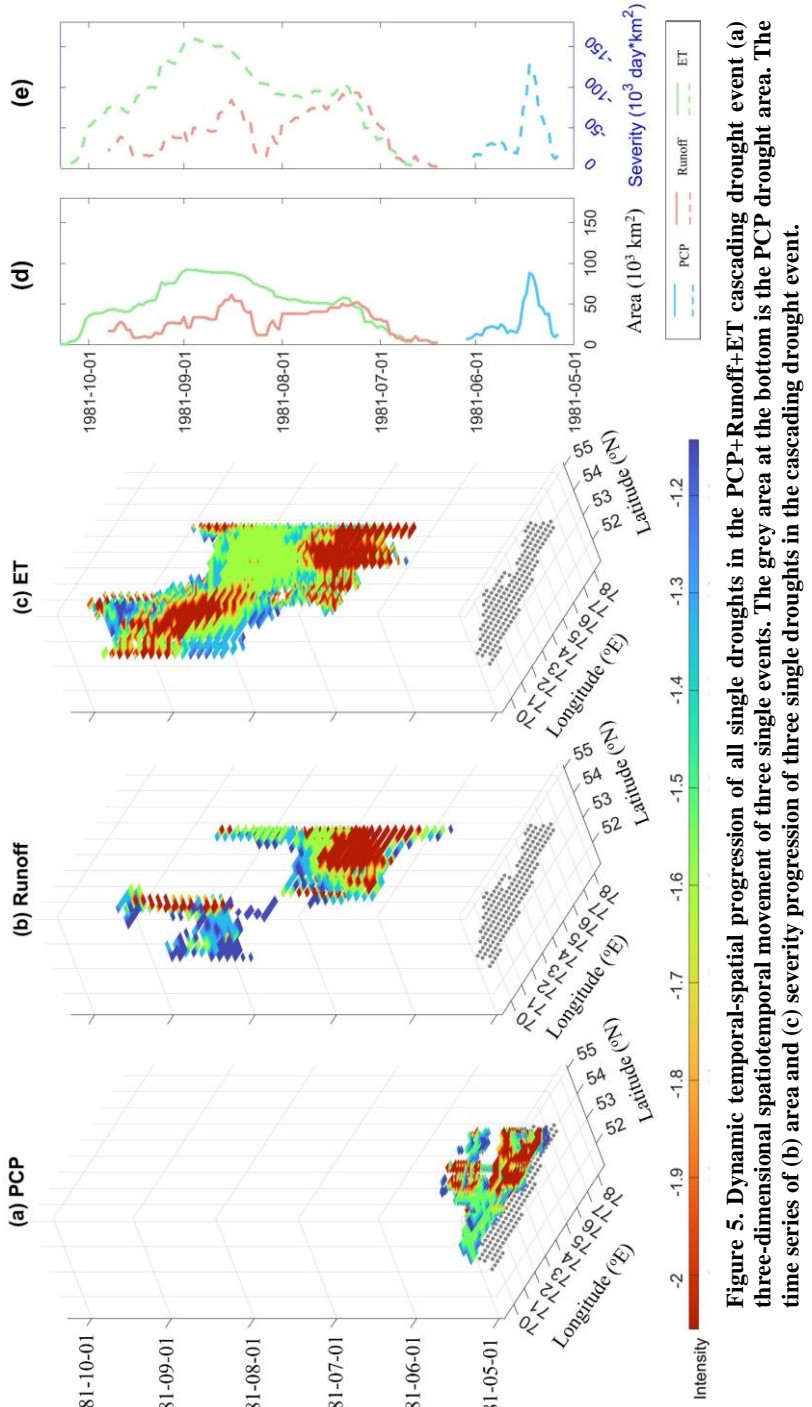

**Figure 5. Dynamic temporal-spatial progression of all single droughts in the PCP+Runoff+ET cascading drought event (a) three-dimensional spatiotemporal movement of three single events. The grey area at the bottom is the PCP drought area. The time series of (b) area and (c) severity progression of three single droughts in the cascading drought event.**





A PCP+Runoff+ET cascading drought event is taken as an example to elaborate the spatial-temporal continuum movement during cascading processes in detail (Fig.5a-c). This cascading event lasted five months, from the onset of the PCP drought on 06-May-1981 to the end of the ET drought on 10-Oct-1981. The affected area spread around the northern region of Kazakhstan, where all single droughts spread east-westwards (Fig.6). The PCP drought developed and decayed rapidly (Fig.5d and 5e) within 30 days. The PCP drought began on 06-May-1981, extended quickly to the peak (88125 km$^2$) within nine days, and then disappeared within 20 days. Runoff drought and ET drought develop relatively slower than PCP drought. The Runoff drought lasted 105 days, starting on 13-Jun, peaking at 17-Aug (61250 km$^2$), and terminating on the 25-Sep. The ET drought began on 21-Jun, peaked on the 30-Aug (92500 km$^2$), and terminated on the 10-Oct.

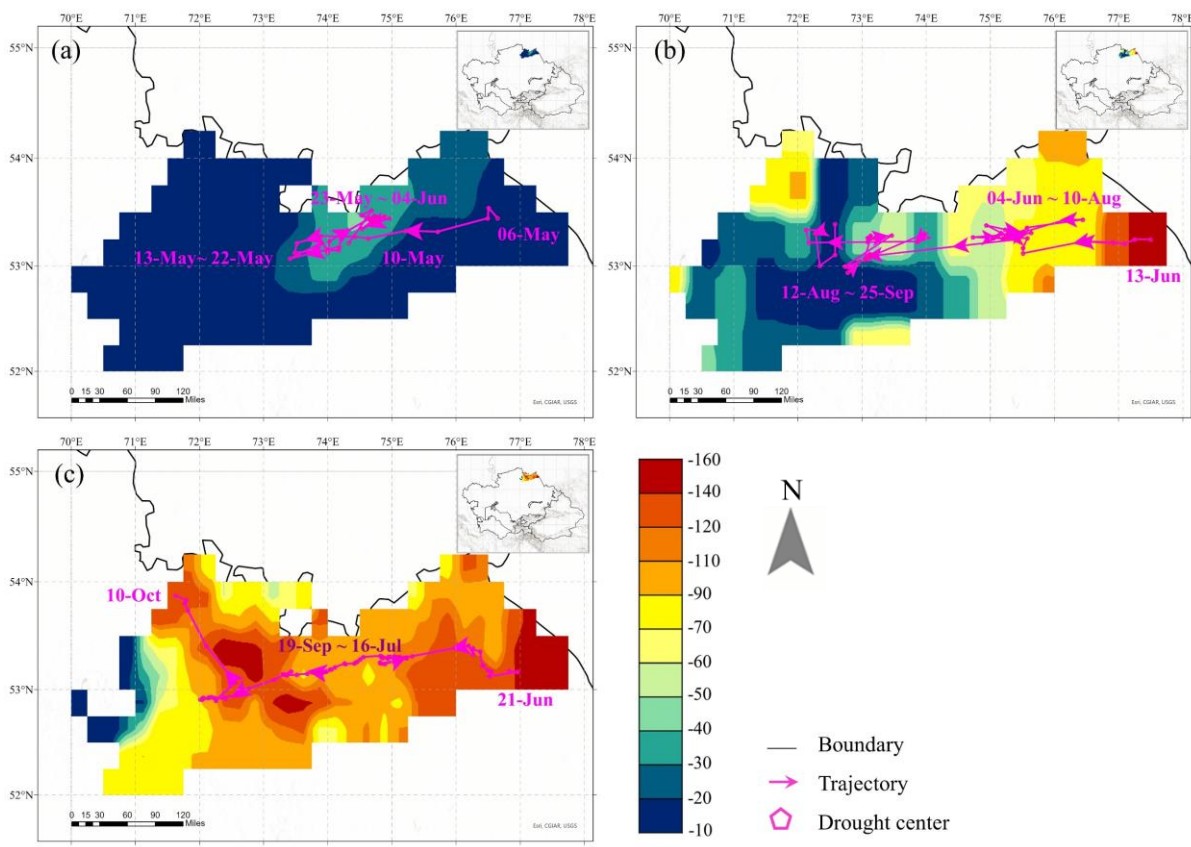

**Figure 6: Cumulative intensity and movement trajectory of the (a) PCP drought, (b) Runoff drought, (c) ET drought in PCP+Ruoff+ET cascading drought event.**



## 4 Discussion

### 4.1 Characteristics of PCP drought could signalize systematic drought risk

Traditional approaches to drought assessment have focused primarily on analysing single systems, neglecting the potential for cascading linkages between different systems, potentially leading to an underestimation of overall drought risk. Some isolated droughts may not be noticeable. However, when multiple hazards are aggregated and occur in conjunction, their combined severity becomes more apparent, leading to more pronounced consequences on the socio-economic system. For example, a single PCP drought of only 30 days, such as the cascading event shown in Section 3.4 (Figs. 5-6), may not be considered a significant drought event. However, when considering the subsequent runoff and ET drought, the cumulative systematic risk of all the droughts included in the cascading drought event must still be considered. Fig 7 shows an apparent increasing trend in total severity as the number of systems increases. CA is dominated by the four-system cascading drought pattern (Fig. 3). This suggests that once the PCP droughts occur, the high systematic drought risk in Central Asia, spreading across the atmosphere, biosphere, hydrosphere and root zone, is likely to emerge. To this end, it is imperative to broaden the scope of drought risk assessments to include cascading hazards to enable the development of comprehensive drought management plans that take into account the magnitude and characteristics of impending disasters (de Brito, 2021).

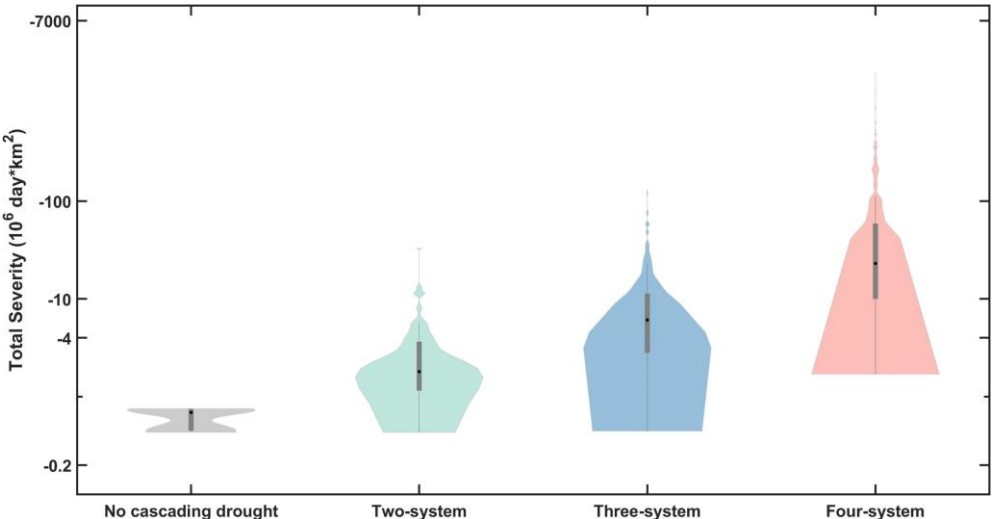

**Figure 7: Violin plot of total severity of cascading drought events in different system cascading patterns.**

As the primary component of the cascading process, precipitation is the most readily available data compared to ET, Runoff, and SM. As such, Fig. 8 proceeds to investigate the features of PCP drought further across various systems of the cascading pattern. Interestingly, the upper threshold of PCP droughts duration in four-system, three-system, two-system, and no cascading drought patterns corresponds to the types of yearly (420 days), seasonal (101 days), monthly (55 days), and sub-monthly (22 days), respectively. This data implies that there would be a four-system cascading drought event after seasonal PCP drought events lasting longer than 101 days occur. The severity and area of in magnitude in PCP droughts increase as the





number of systems increases (Fig. 4), further proved in Fig 8c and 8d. The upper threshold of severity (Fig.8c) in four cascading droughts patterns could reach up to -2022.9, -29.7, -4.7, and -0.76 $10^6$ day*km$^2$, whereases that of area percentage is 99%, 30%, 6.5%, and 1.5%, respectively (Fig.8d)

The intensity of PCP droughts in four-system cascading patterns is higher than in the other three patterns (Fig.8b). However, the intensity of PCP droughts in the no-cascading drought pattern is higher than in the three-system and two-system cascading drought pattern, and in some cases even higher than in the four-system cascading pattern. However, it may be limited by their short duration and small areas. PCP droughts in the no cascading drought pattern did not form the cascading drought event.

The above result highlights that the characteristics of PCP droughts could serve as indicators of systematic drought risk.
Long-term PCP droughts with greater severity and intensity in larger areas are more likely to form extensive cascading droughts across more systems and cause greater systematic risk. Further research is needed to understand the characteristics and underlying mechanisms of moderate and small PCP droughts that trigger cascading drought events with more systems involved.

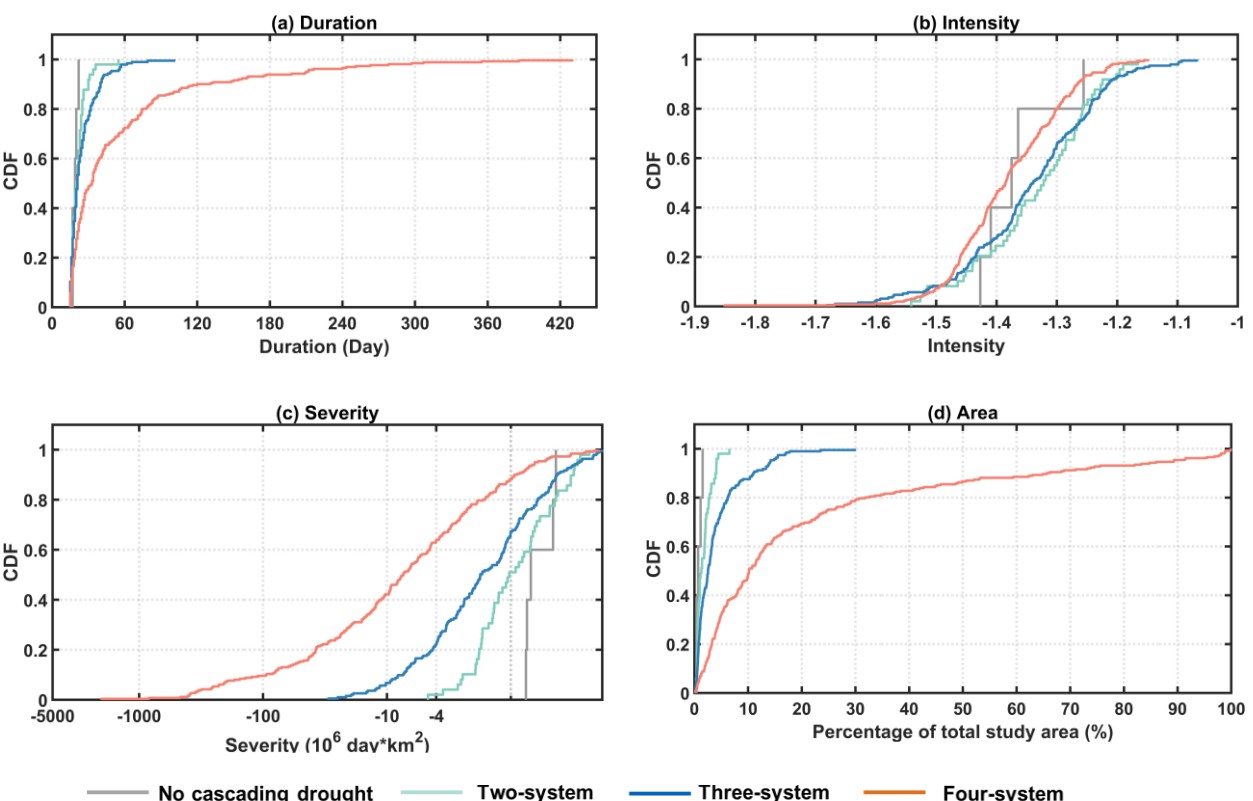

**Figure 8: Cumulative distribution plot of the PCP drought features in the cascading drought pattern: a) duration, b) intensity, c) severity, and d) area.**





### 4.2 The integrated driven effect of energy-limited and water-limited phases on drought progression in CA.

According to the findings presented (Figs. 3 and S1), the initial phase of drought cascades in Central Asia usually involves the occurrence of ET droughts after PCP droughts, followed by runoff droughts. This emergent pattern is the most frequent among the different types of drought cascades studied. The progression from PCP to SM droughts takes the longest time relatively.

Pendergrass et al. (2020) emphasize that the ET drought developing during precipitation deficits can be categorized into energy-limited and water-limited phases. During the energy-limited phase, ET fluctuation is forced by the precipitation deficits as it enhances the evaporative demand. A co-increase in PET and AET would be observed when soil moisture is adequate. When the water-limited phase begins, PET could continue to increase, but the increase in AET is no longer sustained and subsequently decreased, which is restricted by soil moisture deficits. According to Zhao et al. (2022), AET rising beginning from a state of dryness and limited moisture can be triggered by precipitation deficits and enhanced by warm air temperatures. It is more prevalent and often underestimated in relatively arid climates regions dominated by low biomass and water demand ecosystem. Yuan et al. (2023) found that, in regions with higher humidity, such as Europe, North Asia, southern China, eastern and north-western parts of North America, and the Amazon, a precipitation deficit accompanied by an increase in evapotranspiration at energy-limited phase leads to a faster depletion of soil moisture, increasing the probability of flash drought. Mastrotheodoros et al. (2020) also found that annual evapotranspiration (ET) significantly increased during low precipitation in the European Alps, sequentially amplifying the runoff deficit, especially during heatwaves.

In this study, Central Asia is typically characterized by semi-arid and arid climates. It has experienced significant warming in the past years (Mirzabaev, 2013), possibly causing the common occurrence of energy-limited phases. The drought progress to the water-limited phase and ET droughts continue to be amplified. The runoff drought occupies an intermediate position in this sequence because it comprises surface runoff and subsurface runoff flow routes. The slowest moving from PCP droughts to SM droughts is possible because the shrubland and grassland with short-root are the dominant vegetation in the study area (Klein et al., 2012). The sparse vegetation cover results in lower vegetation requirements and less fluctuation of the space-time signal of soil moisture. Another cascading drought pattern also emerges in this study, such as PCP+SM+Runoff+ET, the difference of underlying mechanism hidden in these emergent patterns needs to be further explored in the next step.

### 4.3 The terrestrial system drought persists longer than the drought in atmosphere.

The droughts in terrestrial system, including ET droughts in the biosphere, Runoff droughts in the hydrosphere, and SM droughts in the root-zone soil layer, exhibit a prolonged duration compared to PCP droughts in the atmosphere (Fig. 4). This phenomenon is consistent with the findings of (Jiang et al., 2023), where ecological droughts defined by the drought index including soil moisture and NDVI, persist longer than meteorological droughts defined by SPI. The persistence of ET droughts, Runoff droughts, and SM droughts may be attributed to the fragile ecosystem resilience in Central Asia, which is originally shaped by the arid environment and sparse vegetation cover, leading to desertification risks and poor recovery capability (Jiang et al., 2019). SM droughts have the longest duration, slightest severity, and smallest area relative Runoff droughts and ET



droughts (Fig.3), suggesting that the fluctuations in drought signal become smoother and broader when they proceed from
meteorology through depth to the root zone soil. This result is consistent with the finding of existing research (Farahmand et
al., 2021; Van Loon, 2015). This phenomenon may be driven by the complex trade-offs between vegetation, groundwater, and
human activities that make droughts in terrestrial systems slower, stealthier, and more persistent than droughts in the
atmosphere.

### 4.4  New framework to detect high-dimensional drought cascades

The resemblance of continuously dynamic space-time drought behaviour was employed to examine the cascading connection
between different droughts. It could detect the dynamic space-time cascading connection compared with the previous pointwise
method. Compared to specifying the temporal and spatial overlapping threshold (Jiang et al., 2023; Liu et al., 2019a), the
approach proposed in this study enables us to detect the cascading relationship with or without temporal overlapping (Table 2
and Fig.5). It is more suitable to the actual situation of droughts cascade in the real world as the drought evolving along the
water circle may span long period and interrelated droughts do not necessarily present overlap in time. The similarity rule of
the space-time motion could break the lots of temporal scale and space pre-restrictions in the conventional methodology and
reduce the spatiotemporal incomparability, which is helpful to reduce subjective pre-determination in exploring drought
cascades and proceed in recognizing drought cascades more flexibly. Moreover, beyond the four types of droughts discussed
in this study, the dynamic spatiotemporal movement can be observed within droughts occurring in different systems, such as
the socio-economic system, and other related hazards, such as heatwaves. Consequently, the framework presented in this study
offers valuable prospects to identify multidimensional cascading linkages between different types of hazards.

Conventional studies focused only on the cascading connection between long-term droughts at large scales (Farahmand et
al., 2021). The framework proposed in this paper based on the dynamic space-time motion at the daily time scale enables us
to capture more sub-seasonal droughts at moderate and small spatial scales (Table 2 and Fig.5), such as flash drought. In
addition, the daily drought index could extend the time series length and, to some extent, alleviate the data scarcity constraint
emphasized in previous studies. It provides more input data for analysing temporal and spatial movement correlations (Fig.5)
and more accurate internal spatial trajectories (Fig.6), making it more feasible to proceed with the statistical analysis of cascade
characteristics, especially for sub-seasonal drought. It will provide a better understanding of the evolution of sub-seasonal
drought across multiple systems, which is critical to design more scientific adaptation policies in the face of increased flash
drought and related compound extreme events under human-induced climate change.

Cascading events, in general, refer to the sequential occurrence of different hazards over time. These events can exhibit
correlations from shared drivers, causal mechanisms, or simply by chance. However, distinguishing between these cases poses
a challenge due to limited sample sizes and an incomplete understanding of the intricate system comprising myriad influencing
factors (Zscheischler et al., 2020). This challenge is also present in this study, given the limited knowledge of external
environmental parameters (e.g., temperature, vegetation, and human activities) in the evolution of droughts. In addition, the





cascading connection has proceeded in the spatial area of PCP droughts, and information outside the PCP drought area is not discussed in this work. Furthermore, only one ET/Runoff/SM drought event is identified in each, which may lead to an underestimation of the severity of cascading droughts. Consequently, further research is warranted to explore these aspects in future studies.

**5 Conclusion**

In conclusion, we present a novel daily time-scale framework for detecting cascading droughts across multiple systems in the higher dimension with flexible spatiotemporal overlap, which mainly relies on evaluating the similarity of the dynamic spatiotemporal motion of different types of droughts. This method enables us to recognize cascading drought events incorporating moderate and small drought event (e.g., flash drought). The result exhibits that the four-system cascading drought event is the most common emergent pattern throughout Central Asia. Specifically, ET droughts, Runoff droughts, and SM droughts are more likely to emerge consecutively after the PCP drought, implying the high possibility of the systematic drought risk across the atmosphere, hydrosphere, biosphere, and root zone soil layer throughout CA. The systematic and severe drought risk across multiple systems is more likely to occur when prolonged PCP droughts with high severity/intensity and large spatial extent are observed. Additionally, ET droughts are likely to occur after PCP droughts, often earlier than Runoff droughts, and SM droughts commonly occur at the latest. ET/Runoff/SM droughts exhibit longer duration than PCP droughts, which might be attributed to the poor recovery capacity of the fragile land surface ecosystem of Central Asia. Furthermore, SM droughts at the root zone layer display the slightest severity and smallest area but the most prolonged duration, suggesting that the fluctuations in drought signal become smoother and broader when they proceed from the atmosphere through depth to the root zone soil. Overall, this work manifests more details of spatial-temporal variations within the drought cascades across multiple systems. In the real world, apart from the four droughts mentioned in this study, droughts in other systems and other related hazards, such as heat waves, involve dynamic space-time motion as well. Therefore, the framework proposed here offers a new possibility to detect dynamic droughts cascade between multiple sequential hazards. The adoption of this framework has the potential to facilitate a comprehensive understanding of the intricate dynamics associated with cascading droughts in various elements of the water cycle. Moreover, its practical significance lies in its ability to effectively assess and predict across-system cascading risks in the context of global warming.

**Data availability**

Daily precipitation data were obtained from the APHRODITE dataset (http://aphrodite.st.hirosaki-u.ac.jp/download/), which has a spatial resolution of 0.25°×0.25° and covers the period of 1951-2015. The potential and actual ET derived from the GLEAM v3.5a project (https://www.gleam.eu/) at a spatial resolution of 0.25°×0.25° and the daily time scale, covering the period of 1980-2020. Root soil moisture data were obtained from the simulated result of the Noah model GLDAS, which is at

a spatial resolution of 0.25°×0.25°, 3-hourly temporal resolution and covered the period from 1948 to 2015 (https://disc.gsfc.nasa.gov/datasets/GLDAS_NOAH025_3H_2.0/summary?keywords=GLDAS_NOAH025_3H_2.0). The daily runoff at the spatial resolution 0.5°×0.5° was from the ISIMIP2a project (https://www.isimip.org/protocol/2a/), covering the period of 1971-2010. The software used to generate all the results is MATLAB 2021b.

**Author contribution**

LT: conceptualization, methodology, software, formal analysis, investigation, writing- original draft preparation, writing-reviewing and editing, visualization. MD: resources, writing-reviewing, supervision. JH: data curation, writing-reviewing and editing, funding acquisition, supervision, project administration.

**Acknowledgements**

We acknowledge the financial support from the WE-ACT project funded by the European Commission's Horizon Programme (Grant agreement ID: 101083481). L.T. sincerely thanks Dr. Qing Lin for his support in the writing and coding part of this study. L.T. would like to thank the financial support from the CSC Fellowship.

**Competing interests**

The authors declare that they have no conflict of interest.

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
