# Peer review of "Drought cascades across multiple systems in Central Asia identified based on the dynamic space-time motion approach"

_Hydrology and Earth System Sciences, 2023_

## Author Comment (AC1)

Dear Editor and Reviewer,

Thank you for your letter and the reviewer's comments concerning our manuscript entitled "Drought cascades across multiple systems in Central Asia identified based on the dynamic space-time motion approach" (hess-2023-140). These comments are valuable and helpful for revising and improving our paper and providing significance to our research. We have carefully considered the words and made corresponding responses. The responses to the reviewer's comments are listed below.

**The reviewer's comments are in red text, while the responses are in black. We use the blue text to denote the content that will be added to the manuscript.**

**General comments**

The present paper proposes a novel framework for tracking drought cascades across 10 multiple systems by utilizing dynamic space-time motion similarities. The method seems innovative and the topic interesting. However, due to problems in logic, I would recommend major revision before moving to the next step.

**Response**: Thanks very much for your positive comments on the innovation of this study and for pointing out the logical problems!

**Major comments**

1. The introduction of the method, which is the core of the study, is unclear. For example, are the drought pixels a function of (lat, lon) or (time, lat, lon). How are the drought cells clustered, using which criteria or method. What metric(s) is used to determine similarity? I see some in the paragraph, but the vague description using the text is insufficient in clarification.

**Response**:

Thank you for raising the issue of the study methodology. In order to improve the understanding of the methodology, we will act on the several aspects as below.

- First, the drought pixels were defined by the one-dimensional (1D) function. The drought index value within each pixel is calculated with the time series of precipitation, evapotranspiration, runoff, and soil moisture. We will add the Standardized Drought Analysis Toolbox (SDAT) equation for calculating four types of drought index in section 2.3.1, as below.

$$p(x_i) = \begin{cases} \frac{i-0.44}{n+0.12} & for\ x > 0 \\ \frac{n_0+1}{2(n+1)} & for\ x = 0 \end{cases} \tag{1}$$

i: the ranking position f accumulation value; n: the sample size of all accumulation values within each pixel.

Adjacent drought pixels with common sides were clustered each day to form spatially contiguous 2D drought clusters (latitude and longitude) for each day. Contiguous regions of 3D drought events are tracked over time by looking for overlapping areas between 2D clusters on two consecutive days. If there is an

overlapping area between 2D clusters on two consecutive days, the clusters are considered to be a single 3D drought event.

- Second, the schematic diagram of the methodology (Figure.2) will be updated as follow:

[Figure]

**Figure.2. Schematic diagram of methodology detecting the multiple-system cascading event from the space-time dimension**

- Third, there are two metrics adopted in this study. The first metric is the time series correlation of the area changes of a PCP drought event and its ET/Runoff/SM drought candidates, elaborated in Lines 185-210. The second metric is the difference in the direction of movement between the PCP drought event and its ET/Runoff/SM drought candidates, which was elaborated in Lines 211-218.

We will add Figure R1 in the manuscript to illustrate the second metric.

[Figure]

**Figure R1: The schematic diagram for identifying the motion direction of a single drought event and a cascading event according to the metric of similarity in spatial motion**

2. Loose logic problem. For example, in the method part (section 2.3), the introduction in section 2.3 seems too vague, and does not give you a general idea of the method. It does not give you an idea of the inner-connections between the sub-sections. Also, the section 2.3 has too much sub-sections that is un-necessary, consider combining the sub-sections to make the section more neatly.

**Response**:

Thank you for the comments on the structure of section 2.3. We will address several aspects below to make the section logic more compact and cleaner.

- First, I will generally describe the methodology structure at the beginning of section 2.3 to emphasize the connection between identifying single drought events in section 2.3.1 and cascading drought events in 2.3.2. The original content in the manuscript from Lines 132-139 will be replaced by the below text:

  The approach proposed here is summarised in three parts (Fig.2). The three-dimensional (3D) single drought event is identified in section 2.3.1 based on the continuously dynamic space-time motion. In section 2.3.2, we integrated the different types of single drought events with similar space-time motion to form the cascading drought event. The features of single drought events and cascading drought events focusing on this study were introduced in 2.3.3. As precipitation deficits are generally considered as the initiation of droughts, PCP droughts are considered the commencement of a cascading drought event in this study.

  We will add "single" to the title of section 2.3.1 and revise it to '2.3.1 Identifying single drought events based on the dynamic space-time motion'.

  We will add "single drought event" to the title of 2.3.3 and revise it to '2.3.3 Features of single and cascading drought events.'

- Second, we will combine the sub-section 2.3.1.1 and 2.3.1.2 in section 2.3.1. The updated section 2.3.1 will be elaborated into three parts: 1) drought pixel, 2) drought cluster, and 3) drought event. We will add a description at the beginning of section 2.3.1 to give an overview of defining the 3D single drought event.

  "Here, a single 3D drought event is defined by space-time continuum motion to account for the spatial variations in drought development, the approach including the definition of 1) drought pixels (1D, function of time), 2) drought clusters (2D, function of latitude and longitude), and 3) drought events (3D, function of time, latitude, and longitude). The PCP drought event was identified based on these three steps. Within the area of each PCP drought event, the above three steps were repeated to identify its ET/Runoff/SM drought candidates. The detailed detection approach is as follows."

- Third, in section 2.3.2, the sub-section of 2.3.2.1 and 2.3.2.2 will be combined. Section 2.3.2 will describe corresponding two criteria determining the cascading drought event, namely:

  Criteria 1: Correlation of time-series area changes of the PCP drought event and its ET/Runoff/SM drought candidates.

Criteria 2: Similarity in motion direction between the PCP drought event and its ET/Runoff/SM drought candidates.

3. Confusing subsection in discussion. There's no need to break into too many subsections. Also, it is not clearly to me what the discussions are for? In common discussions, it is usually to raise up an interesting issue with one or more paragraphs. But the discussion here seems to be more apt in the results. Other paragraphs about the issue can be put in discussion.

**Response**: Thanks for pointing out the problems in the discussion! We will combine sections 4.1 and 4.3 to form a new section and move the new section to the result part (Section 3.5) to make the discussion neater. The discussion will focus on two aspects: 1) the driving mechanism behind the cascading drought pattern and 2) the comparison between the approach proposed in this study and previous studies. We will revise and reorganise the discussion section as follows.

First, the title of the new section 3.5 will be revised to "Characteristics of PCP drought could signalize systematic drought risk of cascading events." Section 3.5 emphasizes three following points. To make the structure of Section 3.5 clear, we will add the following three sentences at the beginning of the first, second, and third paragraphs of Section 3.5, respectively.

1) Cascading drought events, which present a systematic drought risk induced by drought evolution, deserve more attention than the traditional focus on individual droughts due to their higher severity (Fig. 7) and the more dynamic spatial and temporal scope.

2) Single droughts in terrestrial systems persist longer than atmosphere droughts (PCP droughts).

3) As the commencement of the cascading drought event, precipitation has the more readily available data compared to ET, Runoff, and SM. To examine if the PCP single drought events could signify the severity of systematic drought risk, features of PCP drought in the cascading events involving the different amounts of systems were investigated in Fig. 8.

Second, the purpose of the original Section 4.2 is to discuss the possible driving mechanism of the cascading drought phenomenon reflected by the different patterns in Central Asia. The title will be revised: "Integrated driving effects of energy-limited and water-limited regimes on cascading drought patterns in CA." The response to the other reviewer's comments presents evidence proving this hypothesis (Figure R1). This section will be rewritten as below:

[revised manuscript text omitted]

The purpose of section 4.4 is to compare and discuss the difference between the previous method and the method developed in this study. Because the reviewer also mentioned this in the fourth comment, this part will be explained in the response to the fourth comment.

4. Is there a comparison with the other drought identification methods/indices? It would be good if the authors are comparing this method to other similar methods or single indices to strengthen the core value of this method.

**Response**: Thanks a lot for the suggestion! We fully agree that it is crucial to compare the method of this study with other methods for similar purposes. To illustrate the difference clearly, section 4.4 (A new framework to detect high-dimensional drought cascades) will be reconstructed as four paragraphs corresponding to the four aspects.

First, we will compare this study with the common pointwise/catchment-wise correlation method. The content in Line 395-397 will be revised as follows:

Many studies are customed to link meteorological drought to hydrological drought by point-to-point time series correlation based on the drought index (Geyaert et al., 2018; Guo et al., 2020; Van Loon et al., 2012). These studies identified their lag, attenuation, prolongation, and pooling based on the time series between two drought types. However, these studies are often conducted at a fixed spatial scale, such as at a gauge station or in a catchment, and the spatial variation in drought progression is not considered.

Second, the difference between the method in this study and the temporal and spatial overlapping method has already been discussed in section 4.4 (Line 397-400). To further clarify this difference, we will add the following sentences at the beginning of this part (Line 397).

More recently, researchers have begun to define three-dimensional single drought events by introducing continuous spatial-temporal motion and, based on this, have adopted temporal and spatial overlap thresholds to determine whether there is a relationship between two drought events (Jiang et al., 2023; Liu et al., 2019). The method proposed in this study not only introduces spatiotemporal motion, but also integrates the time series correlation method. The daily time scale used in this study provides more detailed time series of drought area changes and more accurate spatial trajectories (Fig.7). The high-resolution time series at daily time scale provides more input data for analysing temporal and spatial movement correlations (Fig.6). Based on the above, the method can detect the relationship without temporal overlap (Table 2 and Fig.6), identify the linkage among more than two types of droughts, and address their temporal orders.

Third, since the above two previous approaches identify the relationship between two types of drought events, they do not involve the cascading phenomenon among more than two types of droughts. In the third part, we will compare this work with the previous cascading drought study with multiple types of droughts.

Farahmand et al. (2021) investigated the cascading drought phenomenon involving precipitation, runoff, soil moisture, and streamflow to explore their temporal structure. This study differs from our work in three aspects. 1) This work did not involve identifying the relationship between different drought events, as they directly linked the most significant drought events in a given region to form a cascading event. 2) Moderate and small drought events were also not considered (e.g., flash droughts). 3) They do not take into account area variations and spatial motions during drought progression.

Fourth, the limitation of this study and future work was already discussed in the manuscript (Line 416-424). This part will be kept as original.

**Minor comments**

1.  Line 36, "arising from" -> "arise from".
    **Response**: We will correct it accordingly.
2.  Line 53, "compared to defining the drought…" It is redundant.
    **Response**: Thanks! We will delete it.
3.  Line 72 "Based on above…." It seems weird to say "based on above" in here. The authors seem to have point out the situation in characterizing drought. The logic then is to "fill the gap" rather than to sum up. So it may be better to use "To fill gap.." or similar expression.
    **Response**: We will correct it as "To fill the above-mentioned gaps."

4.  Line 76, "Central Asia (CA) is a typical arid and semi-arid region…..". The sudden jump from discussion of method to introduction of Central Asia in the last paragraph seems weird. It may be better just to say something like "test this method in Central Asia which is a typical arid/semi-arid region". It is better to put more info in the section 2.1 study area.
    **Response**: We will correct it as "This method was applied in Central Asia which is a typical arid/semi-arid region."
5.  Line 82, "The findings of this study are anticipated to ……." The whole sentence may be better in the conclusion section rather than introduction.
    **Response**: Thanks for the suggestion! We will delete this sentence.
6.  Line 83, Please add a section introduction of the sections in the last of the paragraph.
    **Response:** Thanks for the suggestion. The added text is as follows:

    This paper is structured as follows: Section 2 provides a brief overview of the geographic information of CA and describes the datasets used in this work and the procedure for identifying the cascading drought events. The results of the proposed approach and the comprehensive analysis are presented in Sections 3 and 4, respectively. Finally, the conclusions are presented in Section 5.

7. Line 84 "Material & Methods"-> "Data and method".

   **Response**: We will correct it accordingly.

Again, we sincerely appreciate the opportunity to revise our work for consideration for publication in Hydrology and Earth System Sciences. Thank you once again for your valuable comments and suggestions. We sincerely hope that our response will meet with your approval.

Farahmand, A., Reager, J. T., and Madani, N.: Drought Cascade in the Terrestrial Water Cycle: Evidence From Remote Sensing, Geophys. Res. Lett., 48, 10, 2021.

Geyaert, A. I., Veldkamp, T. I. E., and Ward, P. J.: The effect of climate type on timescales of drought propagation in an ensemble of global hydrological models, Hydrol. Earth Syst. Sci., 22, 4649-4665, 2018.

Guo, Y., Huang, S. Z., Huang, Q., Leng, G. Y., Fang, W., Wang, L., and Wang, H.: Propagation thresholds of meteorological drought for triggering hydrological drought at various levels, Sci. Total Environ., 712, 12, 2020.

Hsu, H. and Dirmeyer, P. A.: Soil moisture-evaporation coupling shifts into new gears under increasing $CO_2$, Nat Commun, 14, 2023.

Jiang, T., Su, X., Zhang, G., Zhang, T., and Wu, H.: Estimating propagation probability from meteorological to ecological droughts using a hybrid machine learning copula method, Hydrol. Earth Syst. Sci., 27, 559-576, 2023.

Klein, I., Gessner, U., and Kuenzer, C.: Regional land cover mapping and change detection in Central Asia using MODIS time-series, Applied Geography, 35, 219-234, 2012.

Kurc, S. A. and Small, E. E.: Soil moisture variations and ecosystem-scale fluxes of water and carbon in semiarid grassland and shrubland, Water Resour. Res., 43, 2007.

Liu, Y., Zhu, Y., Ren, L., Singh, V. P., Yong, B., Jiang, S., Yuan, F., and Yang, X.: Understanding the spatiotemporal links between meteorological and hydrological droughts from a three-dimensional perspective, Journal of Geophysical Research: Atmospheres, 124, 3090-3109, 2019.

Mastrotheodoros, T., Pappas, C., Molnar, P., Burlando, P., Manoli, G., Parajka, J., Rigon, R., Szeles, B., Bottazzi, M., Hadjidoukas, P., and Fatichi, S.: More green and less blue water in the Alps during warmer summers, Nature Climate Change, 10, 155-+, 2020.

Mirzabaev, A.: Climate Volatility and Change in Central Asia: Economic Impacts and Adaptation, 2013. 2013.

Pendergrass, A. G., Meehl, G. A., Pulwarty, R., Hobbins, M., Hoell, A., AghaKouchak, A., Bonfils, C. J., Gallant, A. J., Hoerling, M., and Hoffmann, D.: Flash droughts present a new challenge for subseasonal-to-seasonal prediction, Nature Climate Change, 10, 191-199, 2020.

Van Loon, A. F.: Hydrological drought explained, Wiley Interdisciplinary Reviews: Water, 2, 359-392, 2015.

Van Loon, A. F., Van Huijgevoort, M. H. J., and Van Lanen, H. A. J.: Evaluation of drought propagation in an ensemble mean of large-scale hydrological models, Hydrol. Earth Syst. Sci., 16, 4057-4078, 2012.

Yuan, X., Wang, Y., Ji, P., Wu, P., Sheffield, J., and Otkin, J. A.: A global transition to flash droughts under climate change, Science, 380, 187-191, 2023.

Zhao, M., Liu, Y., and Konings, A. G.: Evapotranspiration frequently increases during droughts, Nature Climate Change, 12, 1024-1030, 2022.

---

## Author Comment (AC2)

Dear Editor and Reviewer,

Thank you for your letter and the reviewer's comments concerning our manuscript entitled "Drought cascades across multiple systems in Central Asia identified based on the dynamic space-time motion approach" (hess-2023-140). The comments involve the driven mechanism of cascading drought, which is valuable and helpful for revising and improving our paper and providing significance to our research. We have taken care of the words carefully and made corresponding responses. The responses to the reviewer's comments are listed below.

The reviewer's comments are in red, while the response is in black. We use the blue text to denote the content that will be added to the manuscript.

**General comments**

This study presents a new framework to characterize and track drought events across multiple systems using the spatiotemporal evolution of deficits in precipitation followed by deficits in evapotranspiration, run-off and root-zone soil moisture.

I think this is an important study which will help to better understand the drought occurrence, its evolution and its associated impact on the ecosystem. I do however have some concerns/confusions that I would like to see clarified:

**Response**: Thanks very much for confirming the value of this work!

**Major comments**

1.  The study identifies 503 drought cascading events with multiple combinations within 2-system, 3-system, and 4-system droughts. However, there is little discussion about the expected mechanism behind the occurrence of these drought cascading events mentioned in Table 1 and Figure 3.

**Response**: Thanks for the comment! We agree with the reviewer that the driving mechanism is essential to understanding the drought cascading pattern. In Section 4.2, we will explain the potential driving mechanism behind the cascading drought pattern from a water-limited and energy-limited perspective.

First, we will revise the title of section 4.2 to "The integrated driving effect of energy-limited and water-limited regimes on cascading drought patterns in CA." To clarify the perspective in this section, we will rewrite section 4.2. The following content will be added in section 4.2:

[revised manuscript text omitted]

2. Given the strong coupling between soil-moisture and evaporation (under water-limited regime), why would we expect the two droughts to occur over different periods? The decrease in actual evaporation should be associated with an enhanced water stress so should not we expect the soil-moisture to reduce at the same time?

[Figure]

Figure R1: CDF plot of antecedent root-zone soil moisture 15 days before PCP drought onset

**Response**: Thanks for the comments! We will act on several aspects below to illustrate the reason why the ET and SM droughts could occur over different periods.

First, the water-limited regime are not the only controlling regime in Central Asia. Hsu and Dirmeyer (2023) recognized most of Central Asia as a transitional and energy-limited regime. In addition, an existing study proved that the soil moisture threshold separating water- and energy-limited regimes in nearly all of the diverse ecohydroclimate regions of Africa is around 0.14-0.18 $m^3/m^3$ (Feldman et al., 2019). Figure R1 shows the raw root zone soil moisture 15 days before PCP drought onset. We can see that, under SM drought conditions, over 45 % of the root-zone SM in Central Asia is still above 0.14 $m^3/m^3$.

Other reasons were discussed in the response to comment 1.

3. The study uses evaporative deficit as difference between actual and potential evaporation from GLEAM and the soil-moisture stress from GLDASv2.0-Noah model. How well the GLDAS and GLEAM dataset represent the soil moisture-evaporation coupling?

**Response**: Thanks for the comment! We draw the scatter plot of AET/PET and SM (Figure R2) in all pixels in Central Asia throughout the study period to illustrate this issue. As shown, the root-zone soil moisture does not present a significant linear relation with the actual and potential evaporation, according to the figures—most of the root-zone soil moisture presents stable with the increasing AET and PET.

Three possible reasons might be contributed to this result: 1) First, the SM data adopted in this study are from the soil layer of 40-100 cm. The shrubland and grassland with short roots are the dominant vegetation in the study area (Klein et al., 2012), and their root density is typically concentrated on the 0-40 cm soil layer (Kurc and Small, 2007). The sparse vegetation cover results in lower vegetation water requirements and less fluctuation of the space-time signal of soil moisture in the deep layer. 2) The hybrid effect of water-limited regimes and energy-limited regimes in Central Asia. 3) The overestimation of the reduction in bare soil evaporation during droughts in drier climates (Zhao et al., 2022).

[Figure]

Figure R2: The density scatter plot of coupling soil moisture and potential/ actual ET, respectively. The x-axis of figure a/b is the potential/actual evaporation. The y-axis is the soil moisture. The colour denotes the point density, and the deeper color represents the higher point density.

4. The climatology of Central Asia is described in Figure 1 and section 2.1. But did authors also find any pattern between climatology/landcover-type and different kind of cascading events?

**Response**: Thanks a lot for the suggestion. The effect of climatology-land cover on cascading patterns is a very interesting research topic. The cascading drought patterns in different types of climatology-land cover is our ongoing work (Lu et al., 2022). In the ongoing work, we have categorised the climate regimes of Central Asia into five types: arid desert, arid steppe, temperate, cold, and alpine. We found that the possibility of the cascading phenomenon occurring in temperate is the highest among the five climate regimes. More details will be presented in the following work once published.

5. The study used PCP drought as a commencement for the event. Can the antecedent hydrologic conditions before the PCP event also govern the evolution? For instance, over a wet condition, I would expect Precipitation deficit (PCP) to be followed by energy limited evaporation with increased evaporative demand and that should lead to run-off deficit and eventually drying up the soil so an evaporation deficit i.e., PCP + Runoff + ET. Conversely over a dry surface, ET droughts may be triggered faster due to vegetation-water stress. Also, the precipitation deficits over dry regions could be lower so they may not directly translate into runoff droughts.

**Response**: Thanks for your comments! Antecedent hydrologic conditions are the factors that cannot be ignored in driving the cascading phenomenon. For example, the antecedent wet/dry condition (such as, antecedent soil moisture) could dominate the following response of other physical variables at the catchment/gauge station scale.

However, the wet/dry condition only defined by one parameter is too simple to explain the driving mechanism for the cascading pattern at the continent/global scale. First, the wet/dry condition is challenging to define in the regions containing the complicated climatology-land cover, like Central Asia. Second, one cascading drought event might involve different types of climatology-land cover, and these different kinds of climatology-land cover could interact with each other during the drought evolution. The figure presenting each drought-cascading pattern's antecedent root-zone soil moisture is as below, which does not show significant difference in 15 cascading drought patterns.

[Figure]

Figure R3: The antecedent root-zone soil moisture 15 days before PCP drought onset.

6. High intensity PCP droughts leads to increase in number of cascading system (Figure 8b). Can the intensity of PCP drought may also affect the type of cascading evolution?

**Response**: Thanks for the suggestion. We draw a new figure to compare the intensity of PCP in 15 cascading drought patterns. However, the intensity difference is not significant among different cascading pattern and the result is insufficient to draw a conclusion from this single factor, as it does not take into account other factors such as vegetation type, human activity and the modulator of atmospheric oscillation.

[Figure]

Figure R4: The intensity and severity of PCP drought event in 15 cascading drought patterns.

7. Does the presence of vegetation type over a region affect the pattern of drought cascades? For instance, over region where plants have deep-roots, they may still evaporate during dry periods and may only result in PCP + Runoff type droughts

**Response**: Thanks for the comments! Vegetation type is a critical element in determining cascading drought patterns. The vegetation of Central Asia is multiple and consists of the 'bare area' and 'sparse vegetation' irrigated grassland areas', 'ice and snow', and 'needle-leaved trees'. The class 'needle-leaved trees' in northeast Kazakhstan represents the beginning of the conifer forest of South Siberia (Klein et al., 2012). The vegetation effect on the cascading drought pattern needs to consider vegetation's spatial and temporal variation. First, as the area of each cascading drought event is irregular and might be involved in multiple types of vegetation, spatial vegetation datasets are needed. Second, the cascading drought event often spans a long time across different seasons. The growing time of vegetation varies due to Central Asia's various vegetation types and climate regimes.

Therefore, the integrated effect of vegetation under different climate regimes on cascading drought patterns needs to be explored in detail, which is our ongoing work.

**Minor Comment/Technical corrections:**

- Line 435: "last" instead of "latest"?

**Response**: Thanks for the hint! We will revise it accordingly.

- Given that Violin plots are not very common, it may be helpful for the readers if authors could briefly describe the statistics of the plot before introducing them.

**Response**: Thanks for the hint! A violin plot is an easy-to-read substitute for a box plot that replaces the box shape with a kernel density estimate of the data and optionally overlays the data points. We will add the explanation of violin plot in the Fig.4: "The bandwidth denotes the density of the values quantified by the kernel density estimate. The black circle in the violin plot indicates the median value".

The citation of the violin plot will be added to the manuscript: Bechtold, Bastian, 2016. Violin Plots for Matlab, Github Project https://github.com/bastibe/Violinplot-Matlab, DOI: 10.5281/zenodo.4559847

Thank a lot for all suggested hypothesis in the comment! The PCP characteristics, climatology/land cover type, antecedent hydrological conditions and vegetation type mentioned by the reviewer are essential for exploring the driving mechanism of the cascading drought pattern, which is very helpful for our ongoing work. However, given the spatial and temporal movement introduced in this paper and the complicated climate/terrestrial conditions in Central Asia, the simple comparison or conventional correlation method with single factors may not be appropriate for exploring the driving mechanism of the cascading phenomenon at the large scale. We will conduct a more appropriate framework in our ongoing work to elaborate it.

Feldman, A. F., Gianotti, D. J. S., Trigo, I. F., Salvucci, G. D., and Entekhabi, D.: Satellite-Based Assessment of Land Surface Energy Partitioning-Soil Moisture Relationships and Effects of Confounding Variables, Water Resour. Res., 55, 10657-10677, 2019.

Hsu, H. and Dirmeyer, P. A.: Soil moisture-evaporation coupling shifts into new gears under increasing CO2, Nat Commun, 14, 2023.

Klein, I., Gessner, U., and Kuenzer, C.: Regional land cover mapping and change detection in Central Asia using MODIS time-series, Applied Geography, 35, 219-234, 2012.

Kurc, S. A. and Small, E. E.: Soil moisture variations and ecosystem-scale fluxes of water and carbon in semiarid grassland and shrubland, Water Resour. Res., 43, 2007.

Mastrotheodoros, T., Pappas, C., Molnar, P., Burlando, P., Manoli, G., Parajka, J., Rigon, R., Szeles, B., Bottazzi, M., Hadjidoukas, P., and Fatichi, S.: More green and less blue water in the Alps during warmer summers, Nature Climate Change, 10, 155-+, 2020.

Mirzabaev, A.: Climate Volatility and Change in Central Asia: Economic Impacts and Adaptation, 2013. 2013.

Pendergrass, A. G., Meehl, G. A., Pulwarty, R., Hobbins, M., Hoell, A., AghaKouchak, A., Bonfils, C. J., Gallant, A. J., Hoerling, M., and Hoffmann, D.: Flash droughts present a new challenge for subseasonal-to-seasonal prediction, Nature Climate Change, 10, 191-199, 2020.

Van Loon, A. F.: Hydrological drought explained, Wiley Interdisciplinary Reviews: Water, 2, 359-392, 2015.

Yuan, X., Wang, Y., Ji, P., Wu, P., Sheffield, J., and Otkin, J. A.: A global transition to flash droughts under climate change, Science, 380, 187-191, 2023.

Zhao, M., Liu, Y., and Konings, A. G.: Evapotranspiration frequently increases during droughts, Nature Climate Change, 12, 1024-1030, 2022.